# Exploration of the Potential Transcriptional Regulatory Mechanisms of DNA Methyltransferases and MBD Genes in Petunia Anther Development and Multi-Stress Responses

**DOI:** 10.3390/genes13020314

**Published:** 2022-02-08

**Authors:** Lisha Shi, Huimin Shen, Jiawei Liu, Hongmin Hu, Hongyan Tan, Xiulian Yang, Lianggui Wang, Yuanzheng Yue

**Affiliations:** 1Key Laboratory of Landscape Architecture, College of Landscape Architecture, Nanjing Forestry University, Nanjing 210037, China; sls@njfu.edu.cn (L.S.); shm12345shm@163.com (H.S.); ljw98085@163.com (J.L.); hhm195975@163.com (H.H.); tan15848173205@163.com (H.T.); yangxl339@126.com (X.Y.); wlg@njfu.edu.cn (L.W.); 2Co-Innovation Center for Sustainable Forestry in Southern China, Nanjing Forestry University, Nanjing 210037, China

**Keywords:** *Petunia hybrida*, DNA methylation, anther development, multi-stress treatments

## Abstract

Cytosine-5 DNA methyltransferases (C5-MTases) and methyl-CpG-binding-domain (MBD) genes can be co-expressed. They directly control target gene expression by enhancing their DNA methylation levels in humans; however, the presence of this kind of cooperative relationship in plants has not been determined. A popular garden plant worldwide, petunia (*Petunia hybrida*) is also a model plant in molecular biology. In this study, 9 PhC5-MTase and 11 PhMBD proteins were identified in petunia, and they were categorized into four and six subgroups, respectively, on the basis of phylogenetic analyses. An expression correlation analysis was performed to explore the co-expression relationships between *PhC5-MTases* and *PhMBDs* using RNA-seq data, and 11 *PhC5-MTase/PhMBD* pairs preferentially expressed in anthers were identified as having the most significant correlations (Pearson’s correlation coefficients > 0.9). Remarkably, the stability levels of the *PhC5-MTase* and *PhMBD* pairs significantly decreased in different tissues and organs compared with that in anthers, and most of the selected *PhC5-MTases* and *PhMBDs* responded to the abiotic and hormonal stresses. However, highly correlated expression relationships between most pairs were not observed under different stress conditions, indicating that anther developmental processes are preferentially influenced by the co-expression of *PhC5-MTases* and *PhMBDs*. Interestingly, the nuclear localization genes *PhDRM2* and *PhMBD2* still had higher correlations under GA treatment conditions, implying that they play important roles in the GA-mediated development of petunia. Collectively, our study suggests a regulatory role for DNA methylation by *C5-MTase* and *MBD* genes in petunia anther maturation processes and multi-stress responses, and it provides a framework for the functional characterization of *C5-MTases* and *MBDs* in the future.

## 1. Introduction

DNA methylation is an important epigenetic regulatory feature, and it plays critical roles in multiple plant biology processes by regulating the expression levels of target genes. Additionally, it can be stably inherited by offspring without altering primary DNA sequences [1]. DNA methylation mainly exists in three forms: 5-methylcytosine, 6-methyadenine and 4-methylcytosine, among which the latter two are the predominant forms in prokaryotes, and the former is the prevalent form in plants [2,3]. DNA methylation (5-methylcytosine) usually occurs in three cytosine contexts, CpG, CpHpG and CpHpH, and it is mainly regulated by cytosine-5 DNA methyltransferases (C5-MTases), which can transfer the methyl group from S-adenosyl-l-methionine to the 5 position of the pyrimidine [4]. DOMAINS REARRANGED METHYLASE 2 (DRM2), METHYLTRANSFERASE 1 (MET1), and CHROMOMETHYLASE 3 (CMT3) are three important members of the C5-MTase family, and the establishment of DNA methylation depends on DRM2, whereas MET1 and CMT3 are mainly responsible for maintaining DNA methylation [5].

Methyl-CpG-binding-domain (MBD) proteins can specifically bind to the symmetrically methylated CpG sites of the up-stream regions of the target genes and recruit C5-MTases to silence gene expression by increasing DNA methylation [6]. MBD family members contain a conserved MBD domain, which is composed of approximately 70 aa [7]. The MBD protein family is divided into six subclasses in plants, and it exhibits two evolutionary branches, dicot and monocot, among which subclasses VI and IV are only present in the former, indicating dicot specificity [8,9]. Studies on the MBD family in plants have mainly focused on model plants, such as *Arabidopsis*, rice, maize and populous, which have between 13 and 18 MBD members [9,10,11]. In comparison, knowledge on MBD proteins in ornamental crops is still limited.

C5-MTase and MBD proteins can be cooperatively expressed and regulate many important biological activities in mammals. For example, DNMT1, which is a member of the C5-MTases, cooperates with MBD3 in human to down-regulate TREM2 expression and cause neuronal injury by enhancing the DNA methylation level of the *TREM2 cis*-acting element region [12]. MeCP2 is a member of the human MBD family, and it interacts directly with DNMT1. The resulting MeCP2-DNMT1 complex plays a critical role in genome-wide DNA methylation and affects the differentiation of human dental pulp cells. A similar synergy between MeCP2 and DNMT1 has been confirmed in rat cancer cells [13,14]. However, we find that there is no potential interaction relationship between C5-MTase and MBD proteins in plants by searching in BioGRID [15].

DNA methylation plays an important regulatory role in plant developmental processes, such as leaf growth [16], fruit ripening [17], seed development [18] and vernalization [19]. Remarkably, DNA methylation also participates in anther developmental processes, including tapetum differentiation and tapetum programmed cell death (PCD) [20,21,22]. In *Brassica napus* and *Nicotiana tabacum*, DNA methylation may promote the progress of tapetum PCD. When tapetum cells are active, the DNA methylation level is low; however, as the tapetum enters the degradation stage, the DNA methylation level begins to increase, and reaches its highest level during the tapetum PCD stage [23]. In *Malus*, the level of DNA methylation was found to be increased during the anther developmental process, and highly modified DNA methylation is a characteristic of tapetum differentiation [24]. The genes *NtMET1* and *BnMET1* are members of the C5-MTase family, and they participate in the developmental processes from microspore to early pollen. They are responsible for the increase in DNA methylation during tapetum PCD processes, indicating that MET1 plays an essential role in the maintenance and establishment of DNA methylation during anther development [23]. Thus, DNA methylation plays critical roles in plant growth and anther development. 

In plants, DNA methylation is closely associated with responses to various abiotic stresses, and the expression levels of some C5-MTase and MBD proteins are induced by drought and cold stresses [17,25]. The major plant hormones, such as IAA and GA, also play important roles in defense-related stress responses, but studies on the plant hormone-based regulation of DNA methylation levels are limited [26]. Particularly, GA is also involved in plant growth and differentiation, especially in anther development, which is closely related to male fertility and fertilization [20,21]. The C5-MTase and MBD proteins have been identified in *Arabidopsis thaliana* [11,27], *Zea mays* [25,28], *Solanum lycopersicum* [10,17] and *Triticum aestivum* [29,30]. However, to the best of our knowledge, no research has yet focused on the identification of C5-MTase and MBD proteins in Petunia.

Petunia, which is one of the world’s most famous garden plants, has a long-standing history as a model for molecular biological studies [31,32]. Here, we identified and characterized the C5-MTase and MBD proteins in the whole genome of petunia, including phylogenetic, protein-conserved motif, gene structure and expression pattern analyses. The expression correlations of *PhC5-MTases* and *PhMBDs* in anther developmental processes and different tissues, along with the expression correlations under multiple abiotic and hormonal stresses, were analyzed, and one *PhC5-MTase* and *PhMBD* gene pair that can be co-expressed was selected and subjected to subcellular localization experiments. Our study provides valuable clues to the functional identification and evolutionary relationships of C5-MTase and MBD proteins, and it indicates that they have a synergistic effect in plants.

## 2. Materials and Methods

### 2.1. Identification of C5-MTase and MBD Genes

The members of C5-MTase and MBD families of petunia were identified from the SGN repository database (https://solgenomics.net/organism/Petunia_axillaris/genome, accessed on 8 January 2021). The Hidden Markov model (HMM) profiles of DNA methylase domain (PF00145) and MBD domains (PF01429) were downloaded from the PFAM database (https://pfam.xfam.org/, accessed on 2 January 2021), and HMMER software (version 3.0) was used to scan all the C5-MTase and MBD protein sequences (*E-value* ≤ 1 × 10^−10^) [33]. The web tools Batch CD-search [34] and SMART [35] were used to detect the integrity of the conserved domains of the candidate genes, and genes with incomplete conserved domains were manually removed. The molecular weights and theoretical isoelectric point values of C5-MTase and MBD proteins were calculated using ExPASy (https://web.expasy.org/compute_pi/, accessed on 7 February 2021). The database BioGRID was used to identify interactions between C5-MTase and MBD proteins [15].

### 2.2. Phylogenetic and Gene Structure Analysis

The neighbor-joining phylogenetic trees of C5-MTase and MBD proteins were constructed using MEGA 11 software (bootstrap value = 1000), and the protein sequences of 17 species used in the evolutionary tree were obtained from Phytozome (http://phytozome.jgi.doe.gov/pz/portal.html, accessed on 5 May 2021); they are listed in Appendix A. The sequences of C5-MTase and MBD proteins were visualized and analyzed using Jalview (version 2.0) [36]. MEME was used to identify the preserved motifs [37], and the structures of *C5-MTase* and *MBD* genes were visualized using TBtools software [38]. The gene structures were constructed by IBS 1.0 [39].

### 2.3. Plant Materials and Stress Treatments

Petunia ‘W115’ seedlings were grown under the following conditions in a controlled growth chamber: light/dark of 16/8 h, day/night temperature of 23 °C/22 °C, relative air humidity of 60% and light intensity of 200 μmol m^−2^s^−1^. Anthers were collected when longitudinal lengths of buds reached 0.2, 0.3, 0.5, 1.2 and 3.5 cm. Leaves, roots, stem leaves, petals, and the four whirls of the floral organs (pistil, anther, petal and sepal) were collected at the 0.3 cm flower bud stage and used in the RNA-seq analysis. Petunia ‘W115’ seedlings were selected at 25 days of age for stress treatments, including 10% PEG 6000, cold (4 °C), IAA (100 μmol/L) and GA (100 μmol/L), which were performed in accordance with methods described by a previous study [40]. For the cold treatment, the petunia seedlings were transported to a 4 °C growth chamber. For drought stress, the petunia seedlings were irrigated with a 10% PEG 6000 solution. For hormone treatments, IAA and GA were applied using a spray application method, in which they were sprayed onto the leaf surface until it was sufficiently wet but without droplet condensation [41]. Each stress treatment had three biological replicates. The above-ground tissues of the seedlings were collected at 0, 3 and 24 h. All the samples were immediately frozen in liquid nitrogen and stored at −80 °C for subsequent experiments.

### 2.4. Isolation of RNA, Real-Time PCR, and Gene Expression Analyses

An EASYspin Plus Plant RNA Kit (Aidlab, Beijing, China) was used to extract total RNA from petunia. Total RNA quantity was assayed with the spectrophotometer method (Denovix, DS-11 FX+, Wilmington, DE, USA) for absorbance ratio of 260/280 (1.8 to 2.0). Total RNA was reverse-transcribed using the TransScript One-Step gDNA Remover and cDNA Synthesis SuperMix (Transgen, Beijing, China), following the manufacturer’s protocol at 42 °C for 30 min and 85 °C for 5 s, and was carried out on a Veriti 96 Well Thermal Cycler (Applied Biosystems™, Thermo Fisher Scientific Inc, Waltham, MA, USA). Approximately 5 μg of RNA was used for cDNA was utilized to synthesize the first-strand cDNA. The cDNA was diluted 1:20 with ddH_2_O. Next, 1 μL of the dilution was used in qRT-PCR analyses [42]. The qRT-PCR reaction reagent was prepared using TB Green™ Premix Ex Taq™ (TaKaRa, Kusatsu, Japan). Each reaction (10 μL) consisted of 5 μL TB Green Premix Ex Taq II, 0.4 μL of 10 μM forward and reserve primers, 1 μL of diluted cDNA, 3 μL ddH_2_O, and 0.2 μL ROX reference dye. The qRT-PCR reactions were carried out on a on Step One Plus real-time PCR (Applied Biosystems™, Thermo Fisher Scientific Inc.). The conditions were 95 °C for 30 s, followed by 40 cycles of 95 °C for 5 s and 60 °C for 30 s. Each qRT-PCR analysis had three biological and technical replicates. On the basis of the Ct values of *PhC5-MTase* and *PhMBD* genes, the raw data were quantified using the 2^−ΔΔCt^ method by Excel 2019 [43,44]. The level of *β-actin* transcripts was selected as a reference [45]. All the real-time PCR reactions were carried out on an Applied Biosystems 7500 Real-Time PCR System, according to the manufacture’s instructions. The PCR conditions were 94 °C for 4 min, followed by 32–35 cycles of 94 °C for 30 s, 55–60 °C for 30 s, 72 °C for 1–3 min, and a final extension of 72 °C for 10 min. The qRT-PCR primers were designed using Primer 5.0 and are listed in Appendix A [46]. A heatmap was constructed using the TBtools software with the RPKM data, which was retrieved from the transcriptome data (Appendix A).

### 2.5. Subcellular Localization

The coding sequences of *PhDRM2* and *PhMBD2* without the stop codons were cloned independently into the pCAMBIA Super 1300-GFP vector at the *HindIII* and *KpnI* restriction sites to construct *3*5*S::GFP-PhDRM2/PhMBD2*. For transient expression investigation, tobacco (*Nicotiana benthamiana*) leaves were infiltrated with *Agrobacterium tumefaciens* (GV3101) harboring *35S::GFP-PhDRM2*/*PhMBD2* and a control vector. The infiltrated plants were grown for 3 days before being stained with the DAPI and observed using a LSM710 microscope (Zeiss, Germany) for the green fluorescent protein (GFP) fluorescence signal.

## 3. Results

### 3.1. Identification of PhC5-MTase and PhMBD Genes

To identify *PhC5-MTase* and *PhMBD* genes across the petunia genome, the HMM profiles and the amino acid sequences from *Arabidopsis* were used as simultaneous queries to search for homologous petunia genes (*E*-values < 0.001). In this study, nine *PhC5-MTases* were identified; each contained a complete DNA methylase domain. They were divided into four subfamilies: three CMT, one MET, four DRM and one DNMT (Appendix A). The coding polypeptides of *PhC5-MTases* ranged from 397 (*PhDNMT2*) to 1,557 (*PhMET1*) aa, the theoretical isoelectric points varied from 4.80 (*PhDRM2*) to 6.32 (*PhCMT2*) and the protein molecular weights ranged from 45.22 kDa (*PhDNMT2*) to 174.84 kDa (*PhMET1*) (Appendix A). All the *PhC5-MTase* genes contained conserved DNA methylase domains at the C-termini, but there were different conserved domain combinations at the N-termini among the subfamilies. The MET group members contained two bromo adjacent homology (BAH) domains, but only one BAH domain was observed in CMT group members. The members of the DRM group possessed two UBA domains. Furthermore, C5-MTases that lacked a conserved region at the N-terminal belonged to the DNMT2 group (Appendix A).

In this study, the protein-coding genes containing a conserved MBD domain were considered to be members of MBD family in petunia (Appendix A). After deleting the mispredicted genes, 11 MBD proteins were identified. The molecular weights of the *PhMBDs* varied from 19.70 kDa to 213.81 kDa, and the theoretical isoelectric points ranged from 4.43 kDa to 9.92 kDa. The lengths varied from 167 to 1948 aa, and they were designated as *PhMBD1*-*11* (Appendix A). The petunia genome has not been assembled at the chromosomal level; therefore, the *PhC5-MTases* and *PhMBDs* were named in accordance with the scaffold order.

### 3.2. Phylogenetic Analysis and Classification

To understand the evolutionary relationships and history of *C5-MTase* and *MBD* genes in plants, the full-length protein sequences of 15 (*A. thaliana*, *Brachpodium distachyon*, *Cynara cardunculus*, *Fragaria ananassa*, *Glycine max*, *Oryza sativa*, *P. hybrida*, *Populus trichocarpa*, *Ricinus communis*, *Sorghum bicolor*, *S. lycopersicum*, *Salvia miltiorrhiza*, *Solanum melanogena*, *Solanum tuberosum* and *Z. mays*) and 7 (*A. thaliana*, *Capsicum annuum*, *O. sativa*, *P. hybrida*, *S. lycopersicum*, *S. tuberosum* and *T. aestivum*) species were used to construct the rootless evolutionary trees of the *C5-MTase* and *MBD* genes, respectively. On the basis of the similarities in protein structure and sequence, the evolutionary tree of C5-MTase was divided into four groups, corresponding to the MET, CMT, DNMT2 and DRM subfamilies (Figure 1a). The MBD family was divided into six subclasses (I, II, III, IV, V and VI) on the basis of the sequence similarity within the MBD motif. Each C5-MTase and MBD subfamily was further separated into two groups, dicot and monocot, indicating that the evolution of the *C5-MTase* and *MBD* genes may differ between dicotyledons and monocotyledons (Figure 1b). In terms of the phylogenetic relationships, *PhC5-MTases* and *PhMBD*s were closely related to their orthologs in eggplant, potato, tomato and pepper, suggesting that the *C5-MTase* and *MBD* families may be conserved within Solanaceae.

### 3.3. Multiple-Sequence Alignment and Motif Analysis

To understand the conservation and divergence of the C5-MTase and MBD proteins, we performed a multiple-sequence alignment and analyzed their gene structures and motif compositions using MEME. In total, 15 common motifs of between 28 and 100 aa were predicted in the C5-MTase family (Appendix A). The motif positions of the C5-MTase family members were conserved within each class. The C-terminal DNA methyltransferase domain contained motifs 1, 2, 3, 5, 6, 9 and 13, and there also six highly conserved motifs (I, IV, VI, VIII, IX and X). These six highly conserved sequences were present in all the members of petunia C5-MTase family, which can serve as guides for the identification of potential C5-MTases, and their order of distribution in the four subfamilies differed (Appendix A). Among them, the motifs I and X can bind to AdoMet, and the motif IV, which contains a proline-cysteine doublet, is a functional catalytic site for C5-MTases [47]. Motif VI can provide a glutamate, which is critical for the binding of the target cytosine [48]. Motif VIII can make non-specific contact with cytosine and motif IX is associated with the organization of the target recognition domain (TRD) [28]. The N-terminal domain contained the remaining eight motifs, for the MET subfamily, motifs 10 and 11 represented two RFD domains, that targeting proteins to replication foci in a non-catalytic context [49]. Motifs 2 and 8 represented two BAH domains in the MET subfamily, and motif 14 corresponded to the BAH domain in the CMT family, which links DNA methylation, replication, and transcription [50]. The DRM subfamily contained the unique motifs 3, 7 and 12 (Figure 2a). The exon–intron organization of *PhC5-MTases* was examined further to demonstrate the structural diversity of *PhC5-MTases*. The number of introns in *PhC5-MTases* varied greatly, from 8 to 20, with the majority of *PhC5-MTases* having eight introns (Figure 2a). To confirm the structural properties of C5-MTase and MBD proteins, the conserved domains of *PhC5-MTases* and *PhMBDs* were analyzed. In this study, 15 common motifs were predicted in the MBD proteins (Appendix A). Each MBD protein contained three to six motifs, and the lengths of the motifs were between 18 and 46 aa. The motif positions of *PhMBD*s revealed that all of the MBD proteins contained motif 1, and seven MBD proteins contained motif 5. Motifs 1 and 5 corresponded to the conserved MBD domain that has the function of binding to symmetrically methylated CpG dinucleotides [51] (Figure 2b). The exon–intron patterns of the *PhMBD* family were studied to learn more about the petunia MBD family’s evolution. Except for *PhMBD4*, which has 15 introns, most *PhMBDs* contained one to five introns (Figure 2b).

### 3.4. Expression Analysis of PhC5-MTases and PhMBDs

To investigate the transcriptional abundance levels of all the selected *PhC5-MTase* and *PhMBD* genes, their expression profiles were analyzed using RNA-seq data. The clustered heatmap showed that most of the genes were expressed in all the petunia tissues and showed any specificity in different tissues/stages (Figure 3a,b). Some genes expressed preferentially in anthers, such as *PhDRM2* and *PhMBD2* (Figure 3a,b). Five genes (*PhDRM4*, *PhDNMT2*, *PhDRM1*, *PhMBD7* and *PhMBD8*) had the lowest expression levels in all the tissues. In addition, correlations between the expression levels of *PhC5-MTase* and *PhMBD* genes among the five anther developmental stages and different tissues were explored. The expression levels of the *PhC5-MTase* and *PhMBD* genes were significantly positively correlated in anthers, and the 11 pairs of genes (*PhMBD1* and *PhCMT1/CMT2/DRM3*, *PhDRM2* and *PhMBD2/MBD3/MBD1* and *PhMBD10* and *PhCMT3/MET1/CMT1/CMT2/DRM3*) with the highest Pearson’s correlation coefficient (PCC) were selected (Figure 3d) (PCCs > 0.9, *p* < 0.05). However, the significance levels of *PhC5-MTase* and *PhMBD* generally decreased in different organs and tissues, with the pairs *PhMBD1* and *PhDRM3* (0.91) */PhCMT2* (0.89) /*PhDRM2* (0.85), *PhMBD10* and *PhDRM3* (0.90) /*PhCMT3* (0.89) /*PhMET1* (0.85), *PhMBD2* and *PhDRM2* (0.87) still maintaining significant positive correlations (PCCs > 0.85, *p* < 0.05) (Figure 4).

### 3.5. Expression Analysis of Phc5-Mtases and Phmbds in Response to Abiotic and Hormonal Treatments

It has been reported that the C5-MTase and MBD proteins are induced by drought and cold stresses in plants, and the plant hormones (such as IAA and GA) often play important roles in defense-related stress responses [17,25,26].

To further determine whether the expression levels of the *PhC5-MTase* and *PhMBD* genes were influenced by abiotic and hormonal stresses, the 11 *PhC5-MTase* and *PhMBD* gene pairs, which consisted of five *PhC5-MTases* and five *PhMBDs*, having the highest Pearson’s correlation coefficients in anthers, were examined under four abiotic and hormonal stresses: cold, drought (PEG), IAA and GA treatments. We used qRT-PCR to analyze the gene expression patterns in the petunia seedlings, and the expression levels of all the selected *PhC5-MTase* and *PhMBD* genes were induced by a variety of abiotic and hormonal stresses (Figure 5a and Figure 6a). The expression levels of all the *PhC5-MTase* and *PhMBD* genes were regulated to different degrees under various conditions, and most *PhC5-MTase* and *PhMBD* genes were up-regulated under all four stresses, except *PhCMT2*. The correlations of the *PhC5-MTase* and *PhMBD* gene pairs under multiple stresses were further analyzed, and the correlations between the members of the 11 *PhC5-MTase* and *PhMBD* gene pairs generally decreased under abiotic stresses and hormone treatments (Figure 5b and Figure 6b). Remarkably, some gene pairs maintained significant positive correlations under one or more stresses. *PhMBD1* and *PhCMT1* maintained a positive correlation (0.92) under cold stress but not under drought stress. *PhMBD1* and *PhDRM3* maintained a significant positive correlation (0.85) under IAA-treatment conditions, whereas significant positive correlations, such as *PhDRM2* and *PhMBD2* (0.85) were observed under GA treatment conditions.

### 3.6. Subcellular Localization of PhDRM2 and PhMBD2

*PhDRM2* and *PhMBD2* are preferentially expressed in anthers, and they are significantly positively correlated in anthers (0.92) and all tissues (0.87), as well as under GA-treatment conditions (0.85). To explore the potential functions of the *PhDRM2* and *PhMBD2* genes, *Agrobacterium* cultures containing *35S::GFP-PhDRM2* and *35S::GFP-PhMBD2* constructs were injected into the tobacco leaves for subcellular localization studies. As shown in Figure 7, similar to the empty *35S::GFP*, the GFP fluorescence of *35S::GFP-PhDRM2* and *35S::GFP-PhMBD2* were both found in the nucleus.

## 4. Discussion

DNA methylation is a common phenomenon in plants, which can lead to gene and transposon silencing, and also participates in plant biological processes, which affects many aspects of plant growth and development [1]. As the main regulators of DNA methylation and the interpreter of DNA methylation sites, C5-MTase and MBD proteins, respectively, affect the development of plants by regulating the methylation level and participating in responses to multiple stresses [8,52], and have been identified in maize [25,28] and tomato [10,17]. Petunia is one of the world’s best known ornamental plants and a model for molecular biology research [31]. However, C5-Mtase and MBD proteins have not been identified in petunia. 

In the present study, 9 *PhC5-MTases* and 11 *PhMBDs* were found in the petunia genome, and the multiple-sequence alignment showed that each of the *PhC5-MTase* contained a complete DNA methylase domain at the C-terminus (Appendix A), and all the *PhMBD* genes contained an MBD domain (Appendix A), indicating that they had the typical characteristics of C5-MTase and MBD family members, respectively. Interestingly, despite the genome size of petunia (1.4 GB) being almost 1.5 times greater than that of tomato (900 MB), 1.6 times greater than that of potato (844 MB), 3 times greater than that of rice (466 MB) and 11 times greater than that of *Arabidopsis* (125 MB), the number of C5-MTase members in petunia is very similar, ranging from 7 to 11, indicating that there is no strong correlation between the number of C5-MTase members and genome size. However, unlike the *PhC5-MTases*, the number of MBD genes in petunia was less than in tomato (18), potato (15), rice (17) and *Arabidopsis* (14), and the number of DNA methylation-related genes is associated with genome-wide duplication and gene loss events [10,25,53]. Consequently, we suspect that gene duplication and loss events during evolution may have contributed to the difference in the number of *MBD* genes, which is consistent with the research on maize [25]. 

On the basis of the phylogenetic tree, both C5-MTase and MBD families were classified into two groups, dicot and monocot (Figure 1), indicating that the evolution of *C5-MTase* and *MBD* genes may differ between dicotyledons and monocotyledons, which is similar to previous research [9]. The *PhC5-MTases* are classified into four subfamilies (Figure 1a), MET, DRM, CMT and DNMT2, which is consistent with the results obtained for *Arabidopsis*, tomato, eggplant and strawberry [17,54,55]. However, there are four MET subfamily genes in *Arabidopsis*, but only one in petunia, suggesting that the MET subfamily of genes were lost during petunia evolution. In addition, most of the *C5-MTase* and *MBD* genes of petunia, potato and tomato in the phylogenetic tree tended to cluster on one branch (Figure 1), indicating that the *C5-MTase* and *MBD* genes are highly homologous and conserved in Solanaceae. 

DNA methylation plays important roles in many aspects of plant developmental processes, such as leaf growth [16], fruit ripening [17] and seed development [18]. DRM1 and DRM2 are members of the C5-MTase family, and *Arabidopsis thaliana drm1drm2* mutants display retarded growth and late flowering phenotypes [56]. MBD11 is a member of the MBD family, and *Arabidopsis* plants with inhibited *AtMBD11* display a phenotype of reduced fertility and abnormal leaf and flower development [11]. Remarkably, DNA methylation also participates in the processes of anther development in *Malus* and tobacco, including tapetum differentiation and tapetum PCD [23,24]. The expression levels of *C5-MTase* and *MBD* genes showed obvious tissue-specificity in tomato and strawberry, respectively, and play different functions in plant growth and development [10,55]. Similarly, we investigated the expression levels of C5-MTase and MBD families in different tissues of petunia, and found that most of the genes were expressed in all the tissues (Figure 3a,b), suggesting that *C5-MTase* and *MBD* genes participate in plant biological processes. Some of the *C5-MTase* and *MBD* genes exhibited floral organ-specific expression, especially in anthers. Furthermore, the expression levels of most *PhC5-MTase* and *PhMBD* genes were significantly positively correlated in anthers, and the 11 pairs of genes with the highest PCCs were selected (Figure 3d) (PCCs > 0.9, *p* < 0.05). However, the correlation coefficients generally decreased in different organs and tissues, with only two gene pairs maintaining significant positive correlations (Figure 4), implying that the co-expression of *PhC5-MTases* and *PhMBDs* preferably influenced anther development.

DNA methylation is regulated by various abiotic stresses, such as cold and drought [57,58]. As the key factors in DNA methylation, *C5-MTase* and *MBD* genes are sensitive to various abiotic stresses in many plants, and their expression levels are greatly altered by cold and drought stresses [10,59]. Here, we evaluated the expression levels of the 11 *PhC5-MTase/PhMBD* pairs with the highest PCCs in anthers in response to abiotic and hormonal treatments (Figure 5a and Figure 6a). Most of the selected *PhC5-MTase* and *PhMBD* genes were induced by multiple stresses, and their expression levels increased under cold and drought stresses (Figure 5a), which is similar to in other plants [10,60]. In addition, plant hormones play important roles in defense responses, but studies on DNA methylation induced by plant hormones are limited [26]. In this study, in response to hormonal treatments, most of the *PhC5-MTase* and *PhMBD* genes were up-regulated under GA-treatment conditions, but some were not significantly changed under IAA-treatment conditions (Figure 6a), implying that the *PhC5-MTase* and *PhMBD* genes were preferentially influenced by GA. Furthermore, the correlations between the members of the 11 pairs of *PhC5-MTase* and *PhMBD* genes generally decreased under abiotic stresses and hormone treatments, with only a few pairs maintaining significant positive correlations (Figure 5b and Figure 6b), implying that the highly correlated expression levels in anthers were unstable under multiple stresses. Remarkably, the two nuclear localization genes (Figure 7), *PhDRM2* and *PhMBD2*, were preferentially expressed during anther development and had a significant positive correlation in anthers (0.92), which was maintained under GA-treatment conditions (0.85) (Figure 6b), implying that they were involved in the regulation of anther development and play important roles in GA-mediated development in petunia.

## 5. Conclusions

In this study, we performed a systematic genome-wide analysis of the *C5-MTase* and *MBD* gene families in petunia. A total of 9 *PhC5-MTase* and 11 *PhMBD* genes were identified and categorized into four and six subgroups, respectively, on the basis of the phylogenetic analyses. The motifs and gene structures of the *PhC5-MTases* and *PhMBDs* within the subfamilies were similar, which validated the predicted classifications. Using the RNA-seq data, 11 pairs of *PhC5-MTases* and *PhMBDs* with the highest PCCs in anthers were identified. Interestingly, although most of the selected pairs had positive correlative relationships in different tissues and organs, the significance levels generally decreased under multi-stress conditions. The qRT-PCR results indicated that the expression of the *PhC5-MTase* and *PhMBD* gene pairs could be induced by multiple abiotic and hormonal stresses; however, the correlative relationships were unstable, with only a few pairs maintaining positive correlations. Remarkably, the *PhDRM2* and *PhMBD2* pair had high correlation values in both anthers and under GA-treatment conditions. They also both localized to the nucleus, implying that they play important roles in the GA-mediated development in petunia. Thus, our study indicates a potential epigenetic regulatory mechanism in petunia that functions through the co-operative expression of *PhC5-MTases* and *PhMBDs*, and the results provide a basis for further research on the biological functions and evolution of *C5-MTase* and *MBD* genes.

## Figures and Tables

**Figure 1 genes-13-00314-f001:**
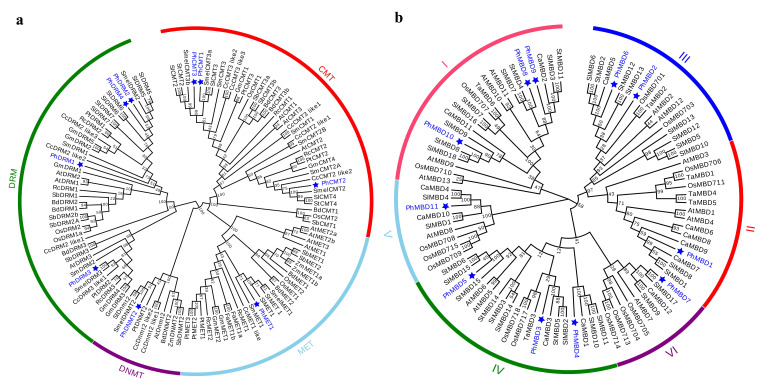
Phylogenetic analysis of the *C5-MTase* (**a**) and *MBD* (**b**) genes. At, *A. thaliana*; Bd, *B. distachyon*; Ca, *C. annuum*; Cc, *C. cardunculus*; Fa, *F. ananassa*; Gm, *G. max*; Os, *O. sativa*; Ph, *P. hybrida* (blue); Pt, *P. trichocarpa*; Rc, *R. communis*; Sb, *S. bicolor*; Sl, *S. lycopersicum*; Sm, *S. miltiorrhiza*; Smel, *S. melanogena*; St, *S. tuberosum*; Ta, *T. aestivum*; Zm, *Z. mays*.

**Figure 2 genes-13-00314-f002:**
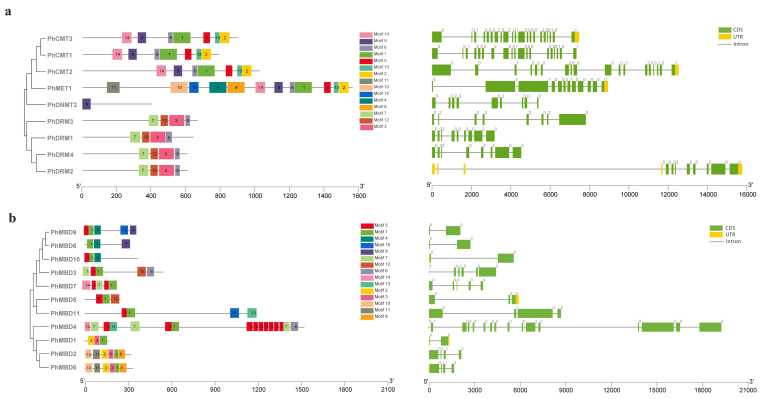
Gene structure and motif analyses of petunia *C5-MTase* (**a**) and *MBD* genes (**b**). The full-length sequences of nine *PhC5-MTases* and 11 *PhMBDs* were used to construct the phylogenetic tree with MEGA 5 software. Lines represent introns, and filled yellow and green boxes represent UTRs and coding sequences, respectively. Different colored boxes indicate the motifs numbered 1–15.

**Figure 3 genes-13-00314-f003:**
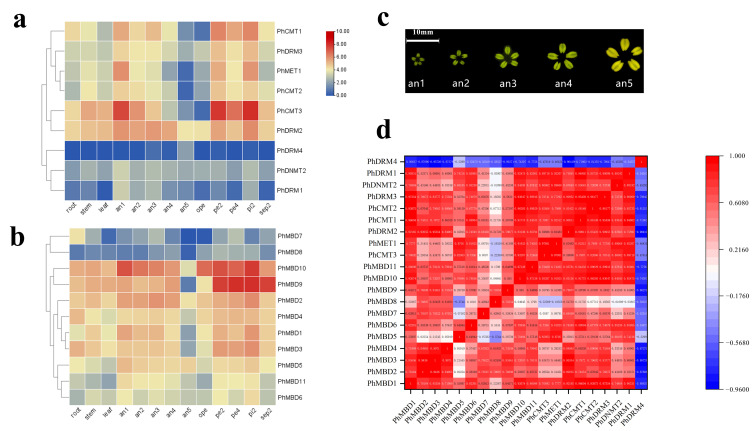
Expression profiles of the identified *PhC5-MTase* and *PhMBD* genes, and the correlations between their expression patterns in anthers. (**a**) The expression patterns of *PhC5-MTases*; (**b**) the expression patterns of *PhMBD* genes; (**c**) a schematic diagram of the five stages of the anther developmental process; (**d**) the correlations between *PhC5-MTase* and *PhMBD* genes in anthers (RPKM > 10). The values in the boxes represent Pearson’s correlation coefficients (PCCs, *p* < 0.05), with red and blue indicating positive and negative correlations, respectively. The labels an1, an2, an3, an4 and an5 represent anthers from buds that reach 0.2, 0.3, 0.5, 1.2 and 3.5 cm in longitudinal length, respectively. The labels pe2, pe4 and ope represent the petals reach 0.3, 1.2 and 3.5 cm in longitudinal length, respectively. The labels pi2 and sep2 represent the pistils and sepals from buds reach 0.3 cm in longitudinal length, respectively.

**Figure 4 genes-13-00314-f004:**
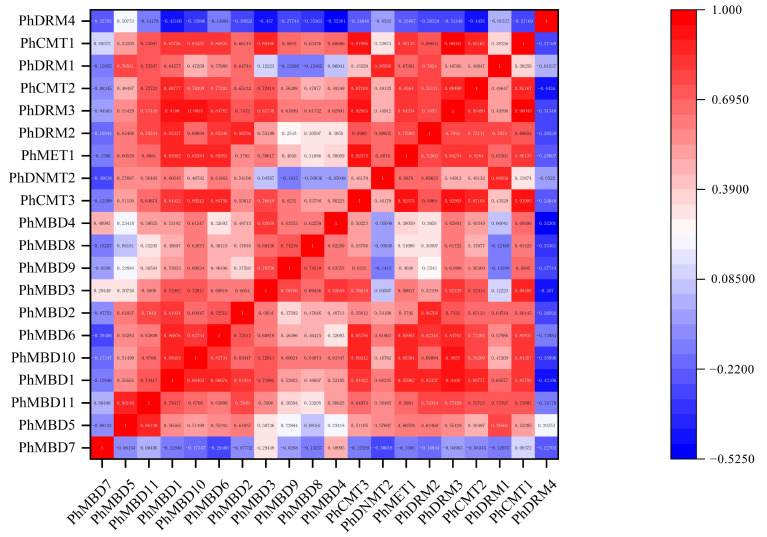
The correlations between *PhC5-MTase* and *PhMBD* gene expression patterns in all the tissues.

**Figure 5 genes-13-00314-f005:**
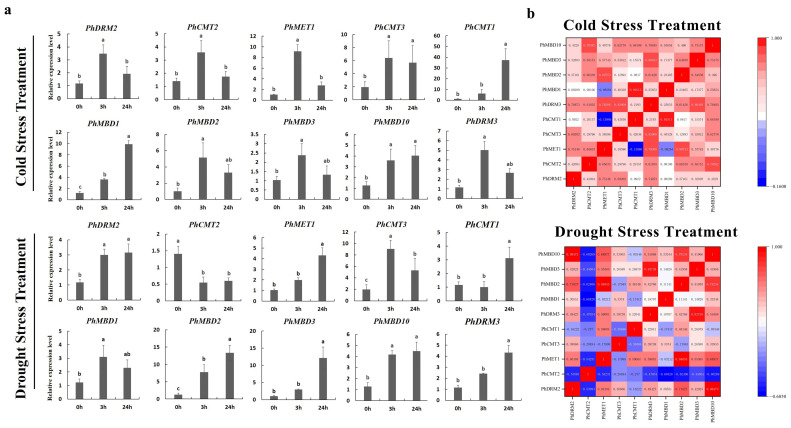
Gene expression levels of six *PhC5-MTase* and four *PhMBD* genes under abiotic stresses (Cold and PEG)**.** (**a**) Expression profiles of the selected *PhC5-MTase* and *PhMBD* genes under cold- and drought-stress conditions; (**b**) the correlations between the selected *PhC5-MTase* and *PhMBD* genes under cold- and drought-stress conditions. The values in the boxes represent Pearson’s correlation coefficients (PCCs, *p* < 0.05), with red and blue indicating positive and negative correlations, respectively. Each qRT-PCR analysis had three biological and technical replicates. Significant differences obtained by Duncan’s test (*p* < 0.05) are indicated by different letters above the error bars.

**Figure 6 genes-13-00314-f006:**
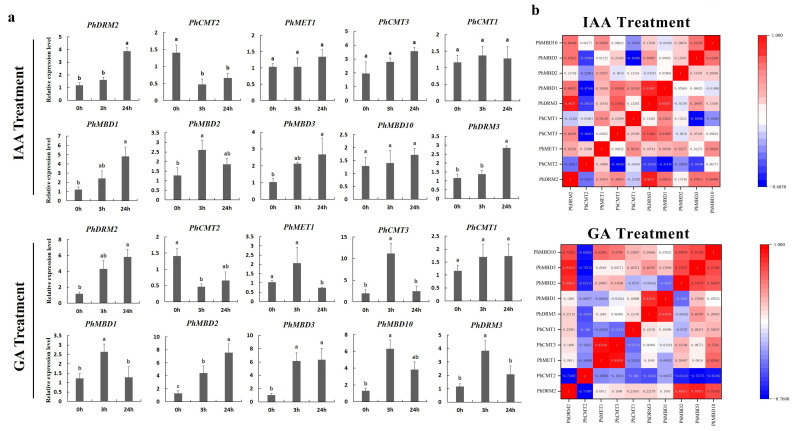
Gene expression levels of six *PhC5-MTase* and four *PhMBD* genes under hormonal stresses (IAA and GA treatments) at the seedling stage. (**a**) Expression profiles of *PhC5-MTase* and *PhMBD* genes under IAA- and GA-treatment conditions; (**b**) the correlations between the selected *PhC5-MTase* and *PhMBD* genes under IAA- and GA-treatment conditions. The values in the boxes represent Pearson’s correlation coefficients (PCCs, *p* < 0.05), with red and blue indicating positive and negative correlations, respectively. Each qRT-PCR analysis had three biological and technical replicates. Significant differences obtained by Duncan’s test (*p* < 0.05) are indicated by different letters above the error bars.

**Figure 7 genes-13-00314-f007:**
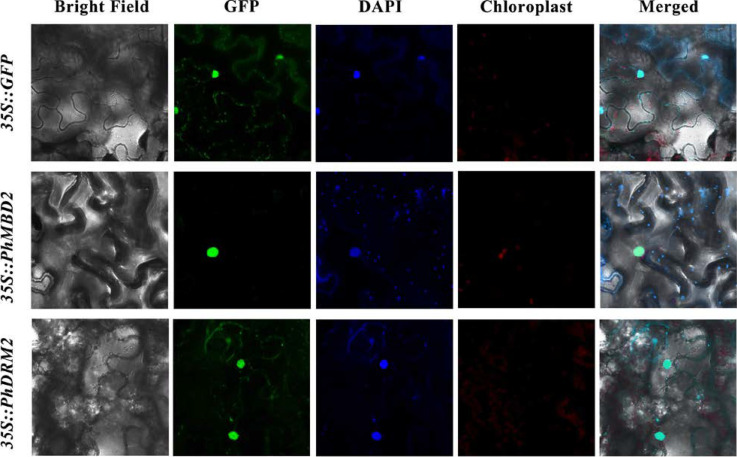
Subcellular localization of *PhDRM2* and *PhMBD2*. Tobacco leaves transformed with *35S::GFP* were used as a control. The green represents the fluorescent signal of the green fluorescent protein (GFP), and blue represents the nucleus visualized by DAPI staining.

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
