# Peer review of "Exploration of the Potential Transcriptional Regulatory Mechanisms of DNA Methyltransferases and MBD Genes in Petunia Anther Development and Multi-Stress Responses"

_genes, 2022, doi:10.3390/genes13020314_

Round 1
Reviewer 1 Report
Authors performed studies on available genomic sequence of Petunia hybrida to found members of cytosine-5 DNA methyltransferases and methyl-CpG-DNA-binding-domain protein genes Research is generally well written, appropriately planned and performed. Conclusions are supported by obtained Results. Some minor comments could be addressed to further improve the manuscript.
Line 49-52
Authors wrote that MBD proteins recruit C5-methylotransferases. Is there a direct or indirect protein-protein interaction between them described earlier? Maybe BioGRID or related database available online (Chatraryamontri et al. 2017) could help to identify such interactions. Authors could add these information to Introduction or Results section. In the last case add appropriate information related to BioGRID database in Materials and Methods section.
Paragraph 2.4
How the quality of isolated RNA was assured?
Amount of RNA/cDNA used for analysis?
Details of RT reaction.
Details of PCR reaction steps, PCR mixture composition.
Equipment used for RT-PCR experiment (name, manufacturer)
Software used for analysis of raw RT-PCR results.
Line 153-154
A link or more detailed information of transcriptomic studies results applied in the research should be provided.
Paragraph 3.3
If possible Authors could add to the Result section the information related to putative biological function/role of sequence motifs found in tested proteins. Maybe they could be associated with posttranslational modifications or play other regulatory role in these proteins.
Author Response
We are very grateful that you have read our work in such a detailed manner and given a positive evaluation, and made some very good questions and suggestions. We have carefully considered each of your comments and make improvements and answers as follows. For more details, please see the attachment.

Reviewer 2 Report
Manuscript ID: genes-1535150
In the present manuscript entitled “Genome-Wide Identification and Expression Analysis of Cyto-sine-5 DNA Methyltransferases and Me-thyl-CpG-Binding-Domain Genes Reveals Their Potential Transcriptional Regulatory Mechanisms in Petunia Anther Development and Multi-Stress Responses” the authors report on the genome-wide analysis of Petunia hybrida in order to identify putative members of the Cytosine-5 DNA methyltransferases (C5-MTases) and methyl-CpG-binding-domain (MBD) genes family; an expression correlation analysis between PhC5-MTases and PhMBDs in anther maturation processes and an expression analysis under abiotic and hormonal stresses conditions.
Minor considerations.
The title of the manuscript is very long and could be reduced.
The data reported in the manuscript is well planned and provides a framework to investigate the putative functional characterization and the possible interaction of C5-MTase and MBD in Petunia hybrida in the future.
The English expose should be revised because there are sentences that require.
Just an example:
line 85-87 ......DNA methylation is closely related to plant responses to various abiotic stresses in plant, and the expression levels of some C5-MTase and MBD proteins are induced by drought and cold stresses in plants.
Author Response
We want to express our deep thanks for your positive evaluation and recognition of our work, each of your comments we have listened carefully and answered as follows. For more details, please see the attachment.
